# Development and Opportunities of Clean Energy in China

**Jin Han \* and Hongmei Chang**

School of Mechanical Engineering, Shaanxi University of Technology, Hanzhong 723001, China; chm130002@snut.edu.cn

\* Correspondence: hanj99@126.com; Tel.: +86-138-9295-1675

**Abstract:** In the context of the energy crisis and global climate deterioration, the sustainable development of clean energy will become a new direction for future energy development. Based on the development process of clean energy in China in the past ten years, this paper expounds on China's clean energy policy and development plan. The development of hydropower, wind power, and solar power in China in recent years is analyzed. On this basis, the Grey Forecasting Model is used to forecast the development and structure of China's clean energy in the next 10 years, point out the direction and market opportunities of China's clean energy development in the future, and put forward the implementation methods for the sustainable development of China's clean energy. It provides a reference for the policy decision-making of China's clean energy development.

**Keywords:** clean energy; hydropower; wind power; solar power; development direction; market opportunities; implementation method

## 1. Introduction

With the increase in global population and the rapid development of the economy, the demand for fossil energy such as oil, coal, and natural gas is constantly increasing. Reducing the development and consumption of fossil energy and protecting the ecological environment is a major problem that must be solved by human beings today. In the past two decades, China's economy has maintained rapid development. At the same time, the rapid growth of energy consumption has also brought about a series of problems such as tight supply and demand and environmental pollution. Therefore, formulating a scientific and rational energy strategy is crucial to China's economic development and environmental improvement. This paper firstly analyzes the distribution, exploitation, and consumption of fossil energy in China in recent years, and points out the importance of developing clean energy for China's energy. Based on the distribution of natural resources such as wind, water, and sunshine, a gray prediction model GM(1.1) is established to predict the development trend in the next 10 years through data modeling of clean energy such as hydropower, wind power, and photovoltaic in China. Analyze the structural changes in China's future energy. Finally, according to China's overall clean energy development goals and opportunities, the benefits of China's clean energy development are summarized.

## 2. World Energy Status

By the 21st century, with the development of the economy, the growth of the global population, and the continuous improvement of living standards, the per capita energy consumption and the total global energy consumption increased rapidly [1]. In 2022, the global demand for natural gas will be around 3.5 trillion $m^3$, and the coal demand will be 80.25 billion tons [2], reaching the highest level in human history.

Although natural energy has guaranteed human life and social development, two very serious problems have arisen due to the continuous increase in human demand for energy. First, the regional shortage of energy is serious [3]. Due to the close relationship between economic development and energy, the regional shortage of energy resources is caused.

Europe, East Asia, South Asia, and Southeast Asia, which are more economically developed or developing rapidly, have become the main regions for global energy import, while some developing countries such as Africa and West Asia have long been faced with insufficient energy production and structural imbalances, resulting in energy shortages. According to statistics, according to the current global human energy demand, the world's existing fossil energy reserves can only sustain human life for about 100 years. Second, the problem of environmental deterioration has become increasingly prominent [4]. For a long time, especially after the first industrial revolution, the development and utilization of a large amount of fossil energy have caused serious environmental pollution [5]. For example, as shown in Figure 1, the continuous increase in carbon dioxide emissions has brought about a series of environmental problems such as global warming, sea-level rise, and frequent outbreaks of extreme weather [6]. With the development of the global economy, the energy demand continues to grow, and solving the environmental problems caused by the development and utilization of fossil energy has become an important proposition faced by all countries. The energy consumption structure dominated by fossil energy is unsustainable [7]. Therefore, the diversified development of energy and the development and utilization of clean energy has received widespread attention from all countries. The "Paris Agreement" [8] states that countries must strengthen the global response to the threat of climate change, control the increase in global average temperature within 2 °C compared with the pre-industrial level, and strive to control it within 1.5 °C.

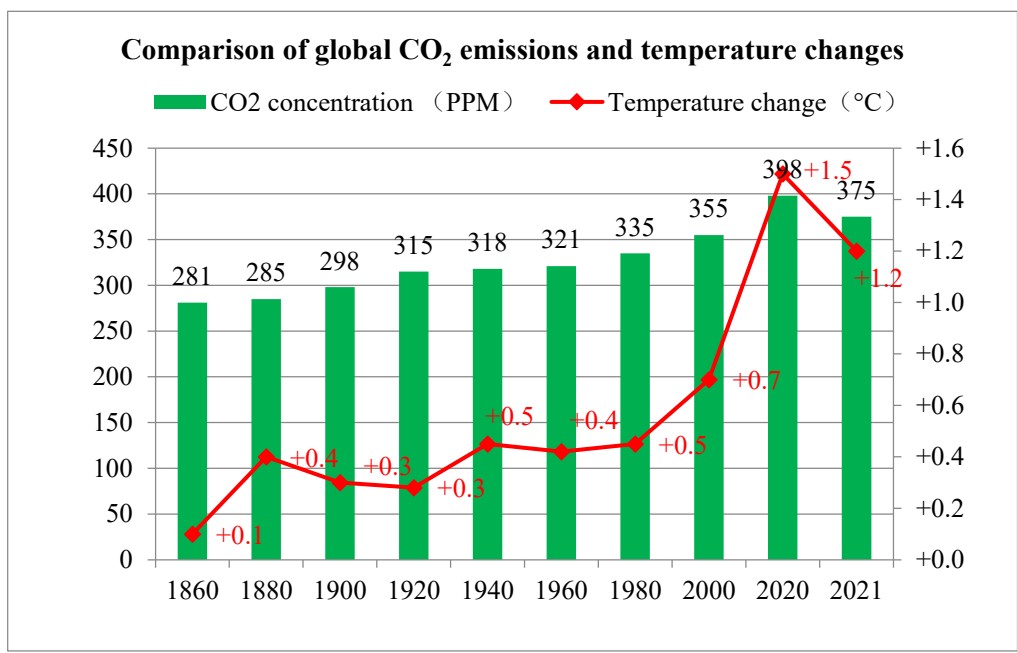

**Figure 1.** Comparison of global $CO_2$ emissions and temperature changes.

### 3. China's Energy Status

China is the country with the largest energy consumption in the world. In the process of rapid economic development in the past 20 years, China's energy demand has increased from 33.3 billion tons in 2009 to 49.3 billion tons in 2020. 2.1% growth rate. In 2021, China's primary energy raw coal output is 4.08 billion tons, a year-on-year increase of 1.4%; crude oil output is 194.77 million tons, a year-on-year increase of 1.6%; natural gas (including coalbed methane, shale gas, coal-to-gas, etc.) output is 192.5 billion cubic meters, a year-on-year increase 9.8% [9]. From the perspective of energy production structure and growth trend, coal has always been the main body of China's primary energy production, accounting for more than 68%, and crude oil production accounts for 6.0%, the production of non-fossil energy such as clean energy is growing rapidly, and the proportion of production is increasing rapidly, and it will be close to 13% [10] of the total primary energy production in

2021 [11]. Although China's output of fossil energy is increasing year by year, China is a country with a population of 1.4 billion, and its total GDP is growing at an annual rate of 6%. According to the 2021 World Energy Statistical Yearbook, although China ranks at the forefront of the world in coal reserves and production, its production rate of oil and natural gas is relatively low and still needs to be imported (see Figure 2). In 2021, China's energy consumption will account for 26.1%, ranking first in the world [12].

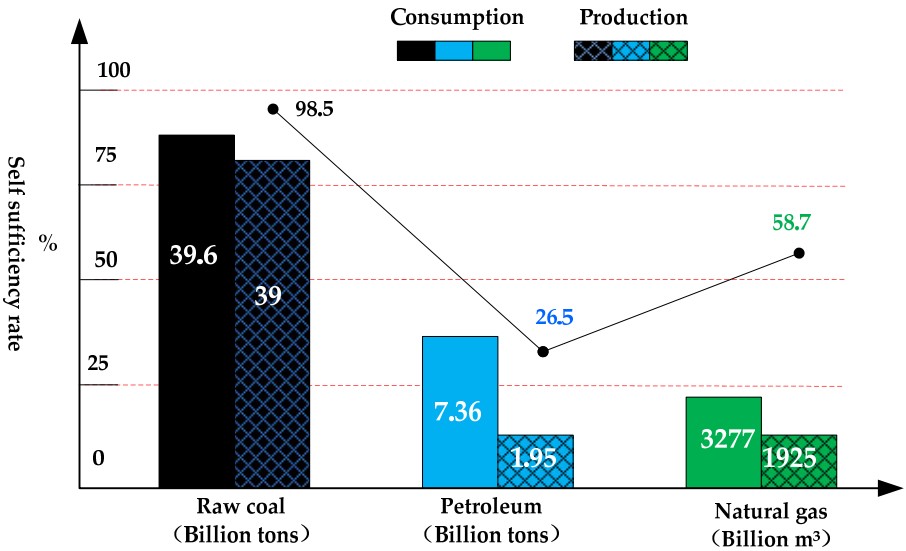

**Figure 2.** Comparison of China's energy output and consumption in 2021 [13].

## 4. Clean Energy in China

At present, different experts and scholars have different understandings of the concept of clean energy. There are mainly these types: one is that clean energy is renewable energy or green energy; the other is that clean energy is a technology system that maximizes the clean development and utilization of energy; the third is that clean energy refers to the energy that does not produce pollution in the process of production and use [14]. In short, the understanding of the concept of clean energy is mainly divided into two viewpoints: one is the energy that does not produce pollution in the process of development and utilization, that is, the energy itself is clean; the second is to develop a technical system for energy utilization, and to make energy clean through technical treatment. According to the understanding of experts and scholars on clean energy, the accurate definition of clean energy should be a technical system of clean, efficient, and systematic application of energy. Clean energy should have the following three points: first, clean energy is not a simple classification of energy, but refers to the technical system of energy utilization; second, clean energy not only emphasizes cleanliness but also emphasizes economy; third, clean energy refers to meeting certain emission standards [15]. China's "Modern Chinese Dictionary (Seventh Edition)" defines clean energy as energy that does not produce or rarely produces pollutants in the process of development and utilization.

### 4.1. The Necessity of Developing Clean Energy in China

With the development of the global economy, the energy demand continues to grow, resulting in the continuous deterioration of the climate and environment and the frequent occurrence of natural disasters. Therefore, solving the environmental problems caused by the development and utilization of fossil energy has become an important proposition faced by all countries. Fossil energy such as petroleum currently stored in the world is expected to be used for only 40 years. As can be seen from Figure 2, China is a big oil-poor country, and its oil and natural gas mainly depend on imports. Therefore, the energy consumption structure dominated by fossil energy is unsustainable [16]. At present, the diversified development of energy and the development and utilization of clean energy

has received widespread attention from all countries. Control greenhouse gases, especially control and reduce carbon dioxide emissions, which account for the majority of greenhouse gases, and finally, achieve a balance between carbon dioxide emissions and removals in the cycle. Achieving carbon neutrality or net-zero carbon dioxide emissions has become the common goal of all countries to combat the deterioration of the global climate [17]. Many countries and regions around the world have formulated emission reduction roadmaps and related laws and signed relevant legal documents under the framework of the United Nations, clarifying the responsibility for addressing climate deterioration and greenhouse gas emission reduction [18] (see Table 1). Countries and regions have successively announced that they will achieve carbon neutrality around the middle of the 21st century.

**Table 1.** List of legal documents concerning the UN and some other countries dealing with the climate change.

| Made by | Name | Time | Reduction Targets |
| --- | --- | --- | --- |
| U.N. | (United Nations Framework Convention on Climate Change) | 1922.05 | Industrial countries reduced greenhouse gas emissions to 1990 levels in 2000 to support climate change activities in developing countries. |
| | (Kyoto protocol) | 1997.12 | By 2010, emissions of six greenhouse gases, such as carbon dioxide, will be 5.2% lower in all developed countries than in 1990. Developing countries have no emission reduction obligations. |
| | (Paris Agreement) | 2015.12 | Controlling the global temperature rise within 2 degrees Celsius compared with the pre-industrial period; And strive for an increase in temperature Limit to 1.5 degrees Celsius. Developed countries continue to be the first to reduce emissions and provide financial support to developing countries. |
| European Union | (European Green Deal) | 2019.12 | The EU greenhouse gas emissions in 2030 dropped by 50% compared with 1990, to 55%, and carbon neutralization in 2050. |
| | (EU climate law) | 2020.10 | Incorporate legislation to achieve carbon neutralization goals by 2050. |
| Japan | (Basic Act on Global Warming Strategies) | 2010.03 | Greenhouse gas emissions in 2020 are 25% lower than in 1990 and 80% lower than in 1990 by 2050. |
| Britain | (The climate change bill) | 2019.06 | Zero greenhouse gas emissions by 2050. |

In recent years, China's carbon emissions have been increasing due to the sharp increase in China's total energy consumption. According to the "bp World Energy Statistics Annual Sign", China's carbon emissions in 2021 will reach 11.9 billion tons, accounting for 33% of the world's total carbon emissions, ranking first in the world [19]. From January to April 2022, China's carbon emissions have reached 4.3 billion tons, a year-on-year increase of 9.8% [20].

*4.2. China's Clean Energy Policy*

To achieve this goal, since 2006, China has promulgated a series of documents on energy conservation, emission reduction, and environmental governance (see Figure 3), continuously promoting energy conservation and emission reduction, continuously optimizing the structure of energy production and consumption, and giving priority to the development of clean energy [21]. Based on China's national conditions, upholding the concept of a community with a shared future for mankind, it actively participates in global climate governance, and has successively signed legal documents such as the "United Nations Framework Convention on Climate Change", "the Kyoto Protocol" and the "Paris Agreement", and proposed the emission reduction target of independent contribution [22].

As the largest developing country in the world, the goal of carbon peaking and carbon neutrality is included in the long-term tasks of China's economic development.

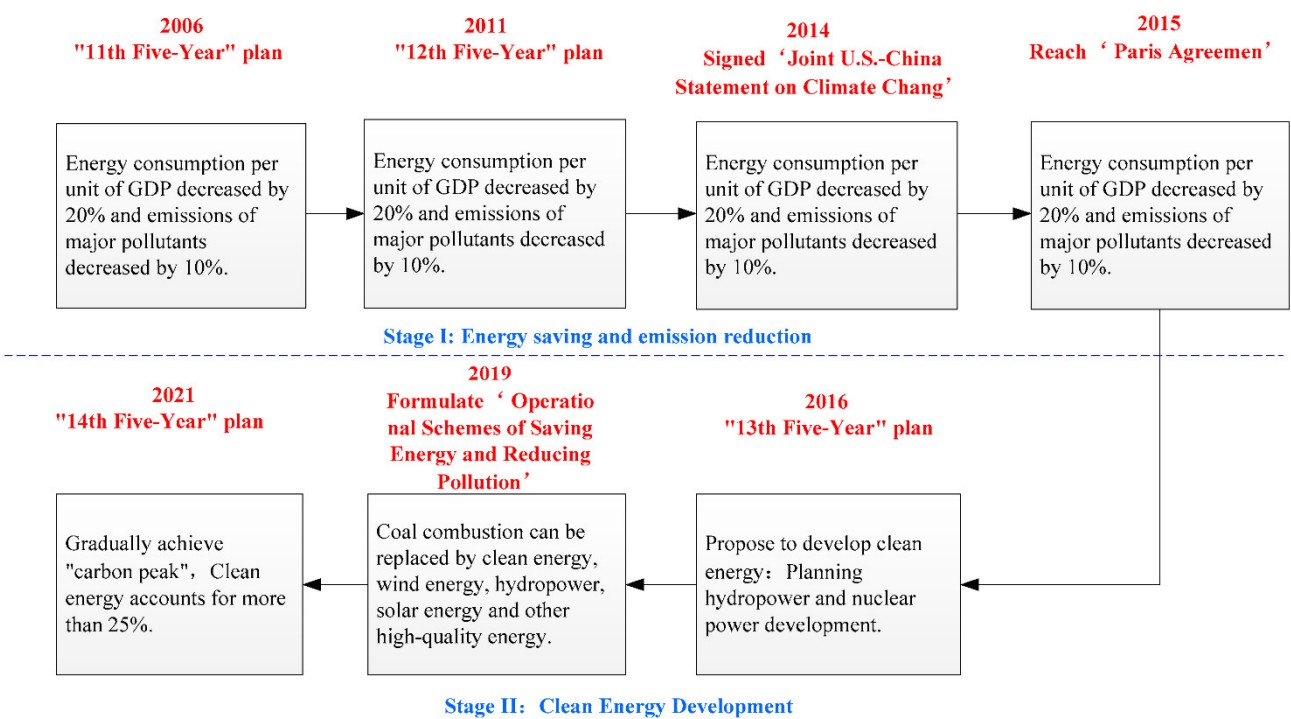

**Figure 3.** China's clean energy policy process.

From Figure 3, it can be seen that the development of China's energy policy is divided into two stages. The first stage is mainly about energy conservation and emission reduction. The largest source of China's energy pollution is coal-fired power generation. As of 2021, China's coal-fired power generation will account for 68% of total power generation [23]. The "Emission Standard of Air Pollutants for Thermal Power Plants" (GB13223-2011) issued by the Ministry of Environmental Protection of China in July 2011 stipulates the emission limits of $SO_2$, NOx, and soot [24], from the emission standards of major air pollutants in major countries in the world (see Table 2). It can be seen that China implements the "ultra-low emission" standard, which is stricter than Japan's emission standard and has become the world's most stringent air pollutant emission standard for thermal power plants. The reason is that, first, thermal power is a major polluter in China's power generation industry. During the 13th Five-Year Plan period, the sulfur dioxide and nitrogen oxide emissions of the thermal power industry accounted for more than 40% of the country's total emissions; secondly, only a quarter of the key cities reached the national standard of secondary ambient air quality level, the number of haze occurrences in some densely populated and economically developed areas remains high. Since 2006, China's air pollutant emissions have dropped significantly, and environmental friendliness has improved significantly. By the end of 2020, 86% of the country's coal-fired power plants have achieved ultra-low emissions [25]. In 2020, China's carbon emission intensity dropped by 18.8% compared with 2015, and 48.4% lower than that of 2005, exceeding the 40–45% commitment to the international community. The target has reversed the rapid growth of carbon dioxide emissions. The second stage is the stage of vigorously developing clean energy, the energy structure will be transformed into green, low-carbon, and intelligent, and the energy-based on solar energy, wind energy, hydropower, etc., and strive to achieve the established goals of peaking carbon emissions by 2030 and carbon neutrality by 2060 [26].

**Table 2.** Emission standard limits of major air pollutants for thermal power plants in major countries in the world [27].

| Nation | Emission Standard Limits (mg/m$^3$) | | |
|---|---|---|---|
| | SO$_2$ | NO$_2$ | SMOKE |
| China | 42 | 100 | 20 |
| USA | 458 | 317.5 | 40.8 |
| Japan | 50 | 100 | 20 |
| EU | 155 | 190 | 30 |

*4.3. Development and Implementation of Clean Energy in China*

With a land area of 9.6 million square kilometers, China has unique natural resources. It has good sunshine conditions in the eastern and northern regions and can develop photovoltaically. The hydropower resources of the 12 western provinces (autonomous regions and municipalities) account for about 30% of the national total. More than 80% of hydropower can be developed; there are abundant wind resources in the northwest and coastal areas, and wind energy can be developed.

4.3.1. Hydroelectric Power

In western China, especially the five provinces (autonomous regions and municipalities directly under the Central Government) of Yunnan, Guizhou, Sichuan, Chongqing, and Tibet in the southwest region account for 2/3 of the national total. Hydropower resources are enriched in the Jinsha River, Yalong River, Dadu River, Lancang River, Wujiang River, Upper Yangtze River, Nanpan River, Hengshui River, Upper Yellow River, Xiangxi, Fujian, Zhejiang, Jiangxi, Northeast, Yellow River, Yuliu and Nujiang River and other hydropower sources [28]. These rivers have concentrated hydropower resources, which is conducive to the realization of cascaded and rolling development of the river basin, the construction of large-scale hydropower energy bases, and the implementation of "West-East Power Transmission" by giving full play to the scale benefits of hydropower resources. The developable installed capacity of China's hydropower resources is about 660 million kW, and the annual power generation is about 3 trillion kWh. From 2015 to 2020, China's hydropower generation has grown steadily (see Figure 4). In 2020, China's conventional hydropower installed capacity has reached 350 million kW, with an annual power generation of 1214.03 billion kWh. By 2050, China's remaining hydropower resources will still be 360 million kW, and the annual power generation will be 190 million kWh [29].

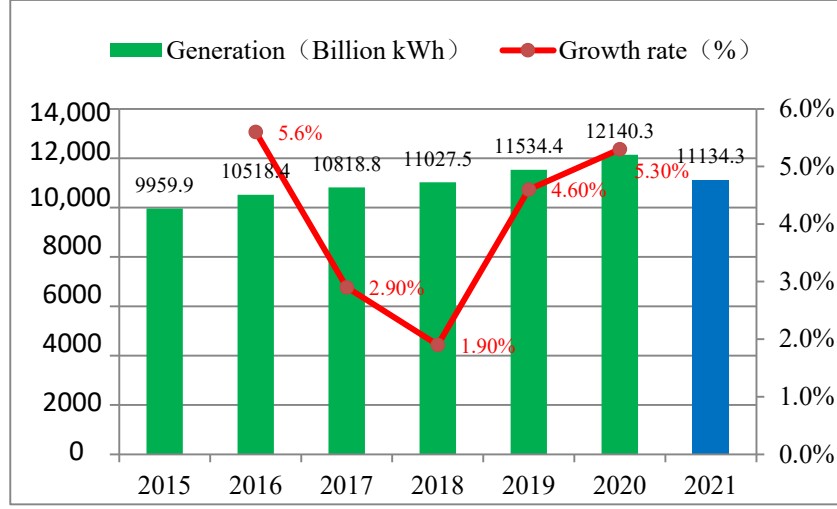

**Figure 4.** China's hydropower generation and growth rate from 2015 to 2021.

### 4.3.2. Photovoltaic Power Generation

China's photovoltaic power generation industry started around 2005 driven by the demand of the European market. It has experienced a development process from scratch to strong in more than ten years and has now become a leader in the world's photovoltaic power generation industry. From 2015 to 2021, China's photovoltaic power generation has increased by nearly 9 times (see Figure 5). In 2021, the total photovoltaic power generation will be 325.9 billion kWh, a year-on-year increase of 24.8.1%; in the whole year, the newly installed capacity of photovoltaic power generation was 54.88 million kilowatts. Provinces with larger installed capacity include Shandong with 4.75 million kW, Hebei with 3.63 million kW, and Anhui with 1.96 million kW [30]. As of the end of December, the national photovoltaic grid-connected installed capacity was 306 million kW, a year-on-year increase of 20.9%.

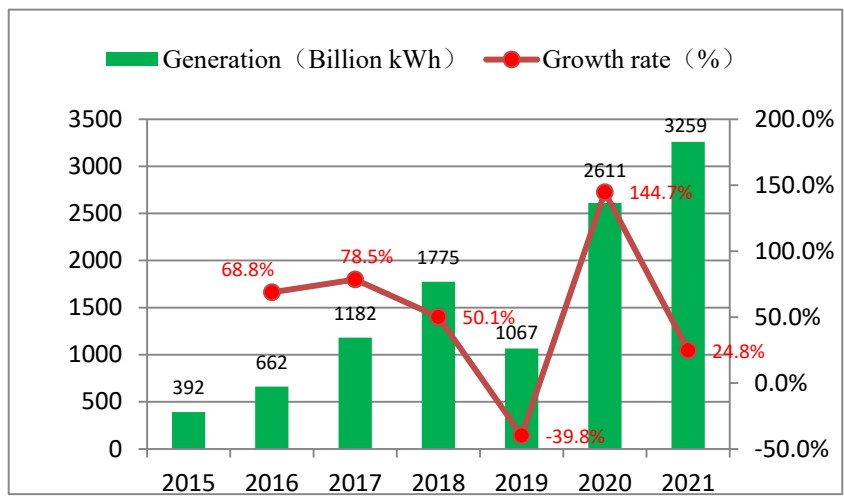

**Figure 5.** China's photovoltaic power generation and growth rate from 2015 to 2021.

Large-scale wind power photovoltaic-based projects focusing on deserts, Gobi, and desert areas will be connected to the grid and put into operation in 2022. The concentration of new energy projects will increase, and the power generation in the west and north will increase rapidly [31]. National strategies such as "numbering the east and counting the west" will also promote the rapid growth of social electricity consumption in the western region and improve the consumption level of new energy power in the region.

### 4.3.3. Wind Power

Wind power generation refers to the power generation method that uses wind turbines to directly convert wind energy into electrical energy. China has listed the wind power industry as one of the national strategic emerging industries. Under the guidance of industrial policies and driven by market demand, China's wind power industry has achieved rapid development and has become one of the few industries in the country that can participate in international competition and gain a leading edge [32].

In the past ten years, China's wind power generation has shown a gradual upward trend, and the growth rate has shown a fluctuating trend. In 2015, China's annual wind power generation totaled 186.3 billion kWh, an increase of 27.6% over the same period of the previous year, the highest growth rate in recent years [33]. In 2020, it reached 414.6 billion kWh (see Figure 6), a year-on-year increase of 15.9%. In 2021, wind power generation reached a record high of 652.63 billion kWh, a year-on-year increase of 57.4%.

In the context of green, low-carbon, and digital transformation, China's wind power generation will usher in a new round of opportunities in the future. The China Business Industry Research Institute predicts that China's wind power generation will reach 9970.79 billion kWh in 2025.

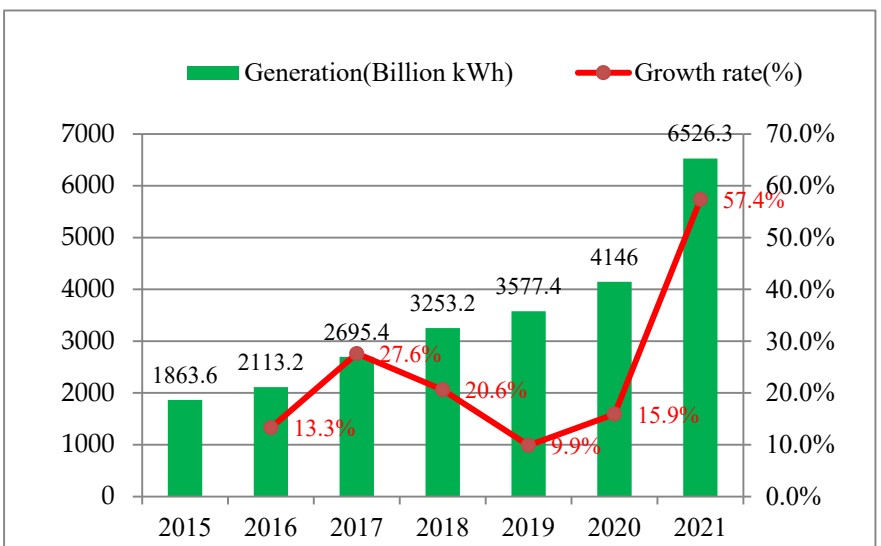

**Figure 6.** China's wind power generation from 2015 to 2021.

### 4.4. China's Clean Energy Trend Forecast

From the current energy situation in China, we can see that the development of China's clean energy is very rapid. According to the current development speed of China's clean energy, China's clean energy will occupy a large proportion in the next 8–10 years. Now by using the GM(1.1) (Gray Forecast Model) [34] model to forecast the total demand for China's hydropower, wind power, photovoltaic power, and electricity in the next 8–10 years, and analyze the structure of China's energy in the future.

### 4.4.1. Establish a GM Prediction Model

Grey prediction is to generate a data sequence with strong regularity from the original processed data and establish a corresponding differential equation, to predict and analyze the future development and change trend of things. GM(1.1) is the most commonly used model in grey system theory. The GM(1.1) model is obtained by generating a new accumulation sequence for the processed original sequence, and further establishing a differential equation through the accumulation sequence. Its advantage is that it only needs a small amount of data to make more accurate predictions, which is suitable for forecasting trends with limited data such as energy and food. The establishment process of the GM(1,1) model is as follows:

**Step 1**: Establish an accumulation sequence and define $x^0$ as the original sequence.

$$x^{(0)} = \left\{ \begin{array}{cccc} x_1^{(0)} & x_2^{(0)} & \ldots\ldots & x_n^{(0)} \end{array} \right\} \tag{1}$$

**Step 2**: The volatility and randomness of the random sequence are weakened by the accumulation of $x$, and the new sequence is obtained as:

$$x^{(1)} = \left\{ \begin{array}{cccc} x_1^{(1)} & x_2^{(1)} & \ldots\ldots & x_n^{(1)} \end{array} \right\} \tag{2}$$

Among,

$$x_i^{(1)} = \left\{ \sum_{j=1}^{i} x_{(j)}^0 \middle| i = 1, 2\ldots\ldots N \right\} \tag{3}$$

**Step3**: Generates a series of equal-weighted neighbors of $x^{(1)}$:

$$z^{(1)} = \left\{ \begin{array}{cccc} z_2^{(1)} & z_3^{(1)} & \ldots\ldots & z_k^{(1)} \end{array} \right\}, k = 2, 3\ldots\ldots n \tag{4}$$

Among

$$z_k^{(1)} = \frac{1}{2}(x_{(k)}^{(1)} + x_{(k+1)}^{(1)}), k = 1, 2 \ldots\ldots n \tag{5}$$

**Step 4**: Create Differential Equations.

$$\frac{dx^{(1)}}{dt} + ax^{(1)} = u \tag{6}$$

Among them, *a* and *u* are the coefficients to be solved, which are called the development coefficient and the gray action, respectively. By calculating the parameters *a* and *u*, $x^{(1)}$ can be obtained, and then the predicted value of $x^{(0)}$ can be obtained. The gray parameter matrix composed of *a* and *u* is:

The discrete prediction formula is: $\hat{a} = \begin{bmatrix} a \\ u \end{bmatrix}$

**Step 5**: Perform mean processing on the accumulated generated data to generate *B* and constant term vector $y_n$

$$y_n = \begin{bmatrix} x_2^{(0)}, & x_3^{(0)}, & \ldots\ldots & x_n^{(0)} \end{bmatrix}^T \tag{7}$$

$$B = \begin{bmatrix} -\frac{1}{2}[x_2^{(1)} + x_1^{(1)}] & 1 \\ -\frac{1}{2}[x_3^{(1)} + x_2^{(1)}] & 1 \\ \ldots\ldots & \ldots\ldots \\ -\frac{1}{2}[x_n^{(1)} + x_{n-1}^{(1)}] & 1 \end{bmatrix} \tag{8}$$

Use the least-squares method to solve for the parameter $\hat{a}$, then

$$\hat{a} = \begin{bmatrix} \hat{a} \\ \hat{u} \end{bmatrix} = (B^T B)^{-1} B^T y_n \tag{9}$$

**Step 6**: Substitute the parameter an into Equation (6), and solve Equation (5) to get the predicted value

$$\hat{x}_{k+1}^{(1)} = [x_1^{(0)} - \frac{\hat{u}}{\hat{a}}]e^{-\hat{a}k} + \frac{\hat{u}}{\hat{a}}, k = 1, 2, \ldots\ldots, n \ldots\ldots 1 \tag{10}$$

**Step 7**: Predicted value validation

$$\varepsilon_k = \frac{x_k^{(0)} - \hat{x}_k^{(0)}}{x_k^{(0)}}, k = 1, 2, \ldots\ldots n \tag{11}$$

When $|\varepsilon_k| < 0.1$, it is considered that the predicted value has reached the highest precision; otherwise, $0.1 \le |\varepsilon_k| < 0.2$, it is considered to have reached a higher precision.

### 4.4.2. Predictive Data Analysis

Table 3 is the statistical data of China's total electricity consumption and clean energy power generation from 2015 to 2021. Bring this data into the GM(1.1) prediction model Formula (8), and get Table 4 China's electricity demand and clean energy from 2022 to 2030 energy generation.

**Table 3.** Statistics of China's Clean Energy Generation (Billion kWh).

| Power Generation | Year | | | | | | |
|---|---|---|---|---|---|---|---|
| | **2015** | **2016** | **2017** | **2018** | **2019** | **2020** | **2021** |
| Total electricity consumption | 55,500 | 59,720 | 63,170 | 66,820 | 70,680 | 74,760 | 79,090 |
| Hydroelectric power | 9959.9 | 10,518.4 | 10,818.8 | 11,027.5 | 11,534.4 | 12,140 | 11,840.2 |
| Photovoltaic power | 392 | 662 | 1182 | 1775 | 1067 | 2611 | 3259 |
| Wind power | 1863.6 | 2133.2 | 2695.4 | 3253.2 | 3577.4 | 4146 | 6526.3 |

**Table 4.** China's clean energy power generation forecast (Billion kWh).

| Power Generation | Year | | | | | | | | |
|---|---|---|---|---|---|---|---|---|---|
| | **2022** | **2023** | **2024** | **2025** | **2026** | **2027** | **2028** | **2029** | **2030** |
| Total electricity consumption forecast | 83,660 | 88,490 | 93,610 | 99,020 | 104,740 | 110,790 | 117,200 | 123,970 | 131,140 |
| Hydroelectric power | 12,456 | 12,808 | 13,169 | 13,541 | 13,923 | 14,316 | 14,720 | 15,135 | 15,562 |
| Photovoltaic power | 4152 | 5531 | 7368 | 9815 | 13,074 | 17,416 | 23,199 | 30,904 | 41,167 |
| Wind power | 7365 | 9193 | 11,475 | 14,324 | 17,880 | 22,319 | 27,859 | 34,775 | 43,409 |

The accuracy of the predicted data is verified by calculating the residual between the predicted data and the original data. It can be seen from Table 5 that the accuracy of the prediction data reaches the highest level in 3 items, and the accuracy in 1 item reaches the higher level. Figure 7 is a predicted fitting diagram of electric energy demand and clean energy generation. It can be seen from the Figure that the predicted value is very close to the actual value, indicating that the overall fit of the curve is good, which shows that the reliability of the prediction result of the GM(1.1) model is very high.

**Table 5.** China's clean energy power generation forecast accuracy.

| Project | $|\varepsilon_k|$ | Precision |
|---|---|---|
| Total electricity consumption | 0.0081 | (<0.1) Highest |
| Hydroelectric power | 0.01 | (<0.1) Highest |
| Photovoltaic power | 0.18 | (>0.1, <0.2) Higher |
| Wind power | 0.077 | (<0.1) Highest |

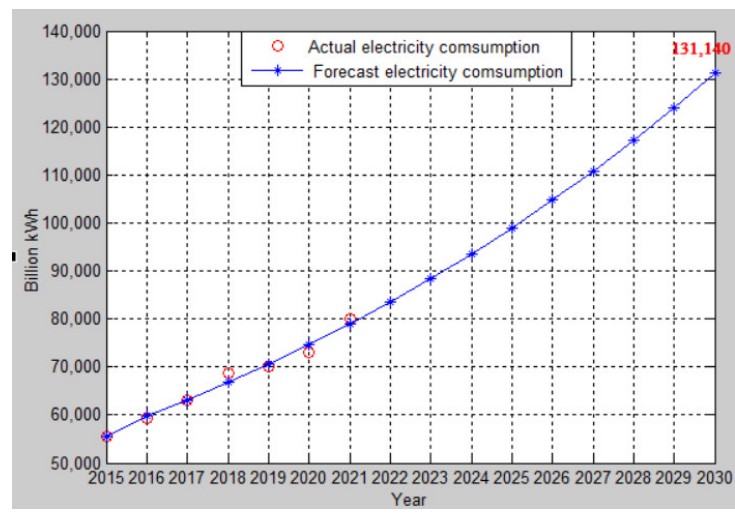

(**a**)

**Figure 7.** *Cont.*

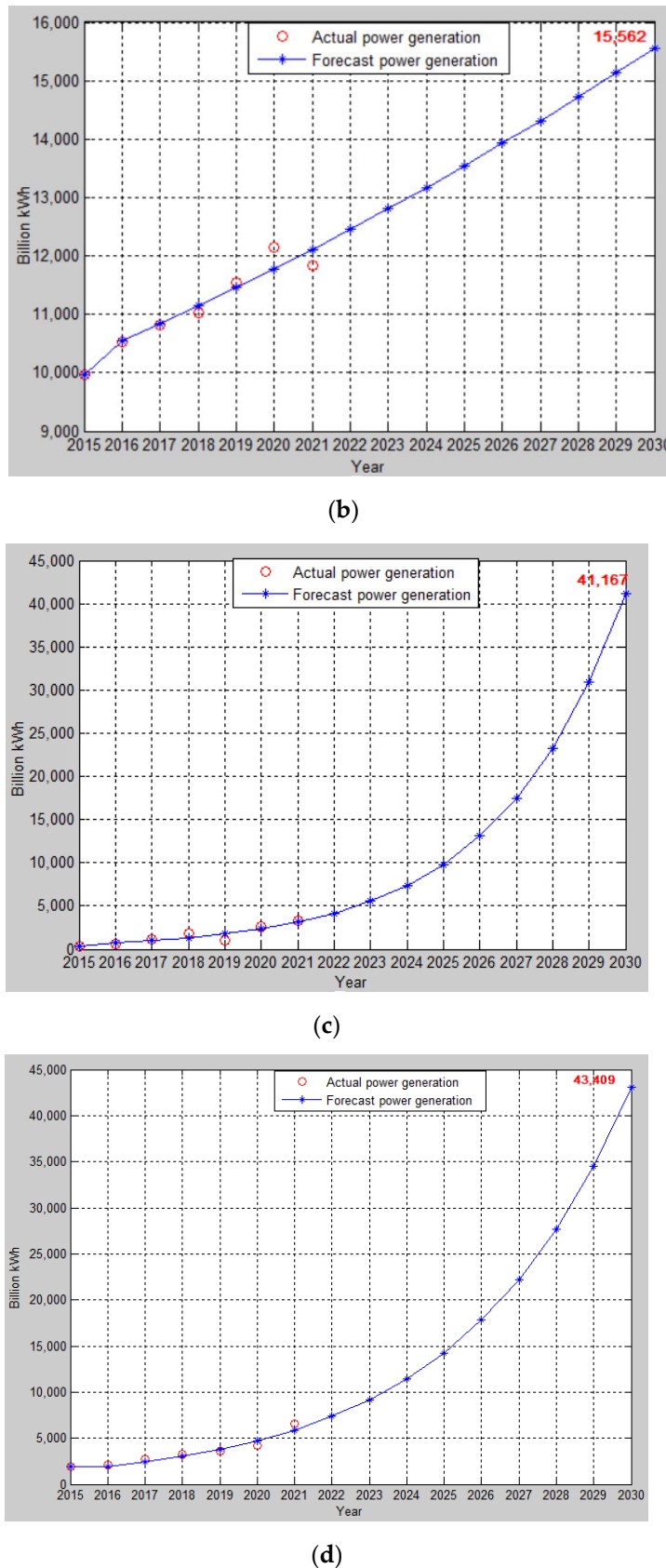

(**b**)

(**c**)

(**d**)

**Figure 7.** China's electricity demand and clean energy generation forecast. (**a**) Electricity demand forecast, (**b**) hydropower forecast, (**c**) PV power forecast, (**d**) wind power forecast.

### 4.5. China's Clean Energy Development Goals

China adheres to the ecological priority, green, low-carbon, and intelligent development path focuses on controlling fossil energy consumption, continues to promote the clean and efficient utilization of fossil energy such as coal, and continuously optimizes the energy consumption structure. It is estimated that by 2030, the proportion of fossil energy in primary energy consumption in China will drop from 84% in 2020 to 35%, and the proportion of clean energy in primary energy consumption will increase from 13% to about 70%. From the predicted value in Figure 7, it can be seen that China's electricity demand in 2030 will be 131,140 billion kWh, and the total clean energy power generation will be 100,138 billion kWh, accounting for 76.36%, which can achieve the expected goal. It is estimated that by 2035, China's fossil energy consumption will drop to 20%, and coal, oil and gas, and clean energy will present a "three pillars" pattern [35]. It is estimated that by 2060, the proportion of fossil energy consumption in primary energy consumption will drop to 15%, and carbon neutrality will be achieved (see Figure 8).

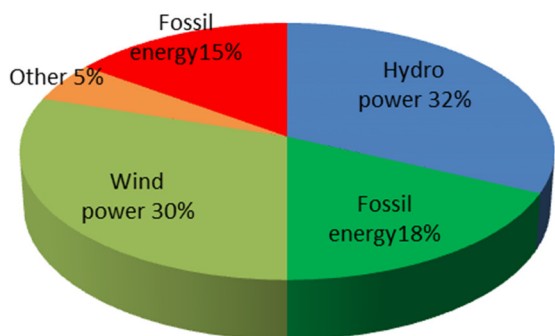

**Figure 8.** 2060 energy consumption forecast.

### 4.6. China's Clean Energy Development Opportunities

- To support the development of clean energy, the government has set up special institutions in the energy and environmental protection departments, has provided tax, price, and other preferential policies for the clean energy industry, and supported the technological innovation and large-scale development of clean energy enterprises [36];
- Fossil energy purification technology is gradually advanced. China ranks first in the world in terms of coal chemical technology, production capacity, and output. China's coal-dominated energy structure has always put enormous pressure on the ecological environment. The development of clean coal technology has transformed the traditional coal industry into a modern coal chemical industry, achieving the effects of energy conservation, emission reduction, and environmental protection;
- The overall goals and layout of all types of clean energy development are clear. Remarkable results have been achieved in clean energy and green development. By the end of 2021, China ranks first in the world in terms of total installed capacity of hydropower, wind power, and solar power. At present, China's biomass power generation has formed a certain scale, showing a good momentum of development;
- The development and utilization of clean energy have improved the quality of life of farmers and the rural environment. China's rural areas have gradually promoted the use of clean energy from coal-gas-electricity combustion, and related projects and facilities have also increased year by year, which has broad market prospects. The use of clean energy has greatly reduced carbon dioxide emissions, greatly improving the ecological environment and the quality of life of residents;
- Vigorously developing the electric vehicle industry, the government encourages the production of electric vehicles and provides subsidies. By 2021, China's electric vehicle market share will exceed 10%.

## 5. Conclusions

This paper puts forward the necessity of developing clean energy by analyzing the current situation of world energy and environmental changes. Aiming at China, the world's largest energy-consuming country, this paper analyzes the structure and demand of China's energy and uses the GM(1.1) forecast model to forecast China's electricity demand and clean energy generation from the future to 2030. The following conclusions can be drawn from the forecast:

- China's clean energy power generation will increase at an annual rate of about 10%, and the proportion of clean energy power generation will reach 76.36% in 2030, reaching the expected goal;
- China's plan to achieve carbon neutrality by 2030 and peak carbon by 2060 is feasible.
- In addition, the development of clean energy in China can not only improve the energy structure but also bring other benefits, mainly in the following two aspects:

**On the social side:**

- The application of clean energy is an effective means to change the way of economic growth. It can reduce pollutants and reduce the operating costs of enterprises, and at the same time improve the economic and social-economic benefits of enterprises;
- The application of clean energy is the best way and an inevitable choice to protect the environment. It can improve the current situation of terminal pollution control. The prevention-oriented concept of environmental protection and development enables enterprises and personnel to consciously use clean energy, eliminate pollutant emissions and production, and achieve better environmental governance benefits;
- The application of clean energy is a major symbol of the construction and development of modern industrial civilization, establishes a green and environmentally friendly corporate image, and promotes the construction of modern industrial civilization in the country.

**On the economic side:**

- Reduce environmental pollution problems and reduce the economic cost of pollution control;
- Reducing the import dependence on traditional energy can prevent the economic impact caused by the price fluctuation of international traditional energy and ensure the economic security of China's energy. It can solve the economic problems caused by the depletion of energy (especially fossil energy). It is conducive to optimizing the national energy allocation structure and improving the economic benefits of comprehensive utilization of energy.

**Author Contributions:** Conceptualization, J.H.; writing—review and editing, J.H.; validation, J.H. and H.C. All authors have read and agreed to the published version of the manuscript.

**Funding:** This research received no external funding.

**Institutional Review Board Statement:** Not applicable.

**Data Availability Statement:** Not applicable.

**Conflicts of Interest:** The authors declare no conflict of interest.

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
