# Peer review of "Development and Opportunities of Clean Energy in China"

_applsci, doi:10.3390/app12094783_

Round 1
Reviewer 1 Report
Comments and suggestions for Authors:
In the paper the authors review China’s energy production from hydropower, wind power generation and solar power generation. It is also discussed development direction and market opportunities for the Chinese energy sector.
The information presented in the paper is interesting to read but I can’t find scientific methods or tools.
The topic considered in the paper needs a deeper description and more analysis with the use of scientific tools and methodology.
I have several specific comments and questions:
- The authors cited relevant paper but I can’t see the authors contribution to the ‘clean energy’ topic. What is the paper novelty? How the authors contributed to the topic?
- Paper title and Abstract – could be improved to better describe the paper scope.
- Keywords – additional keywords could be added.
- The introduction section needs to be improved with a description of the scope of work/objectives and the novelty of this paper.
- Lines 51-56, Figure 2, Table 2 etc – References should be provided for the data that is cited in the paper.
- Line 81 – The ‘clean energy’ definition proposed by authors is ambiguous. Who are the experts and scholars that authors cite for this definition?
- Line 95 – Why the energy consumption structure is unsustainable? If a paper is cited an argument/commentary needs to be provides so the reader understands the subject discussed.
- Table 1 –There is a spelling mistake in the Britain row ‘2050ata’.
- Line 112 – As we are in 2022, the forecast for 2021 is irrelevant. Please provide with up-to-date data in the article.
- Figure 3 – It will be useful to provide a discussion on clean energy targets if were achieved or not and the reason behind this.
- Line 143 – The entire paper needs to be updated to reflect the current year 2022.
- Table 2 – A discussion is needed for Table 2. What is the source of table 2 information?
- The ‘Conclusion’ section should be improved. Please provide specific conclusions from the study conducted.
Author Response
Dear Reviewer:
Thanks for providing us with this great opportunity to submit a revised version of our manuscript. We appreciate the detailed and constructive comments provided by the reviewers. We have carefully revised the manuscript by incorporating all the suggestions by the review panel.
We hope this revised manuscript has addressed your concerns, and look forward to hearing from you.
Your Sincerely,
Jin Han

Reviewer 2 Report
A spell and grammar check is recommended. the plots can be improved especially figure 1 and its caption. figure two looks awkward with the axes reversed. Could also be increased the literature and
Author Response

(The authors gave the same response as above.)

Reviewer 3 Report
The manuscript reported the development process of clean energy in China and expounds on China's achievements in hydropower, wind power generation and solar power generation. This manuscript also analyzes and discusses the development direction and market opportunities of clean energy in China in the future, and puts forward the implementation methods for sustainable development.
I consider the content of this manuscript will definitely meet the reading interests of the readers of the Applied Sciences journal. However, there are certain English spelling and grammar issues, and also the discussion and explanation should be further improved.
Therefore, I suggest giving a minor revision and the authors need to clarify some issues or supply some more experimental data to enrich the content. This could be a comprehensive and meaningful work after revision.
- For the Keywords, ‘hydropower’, ‘wind power’, ‘solar power’, ‘development direction’, and ‘market opportunities’ should also be added to attract a broader readership.
- For grammar issues, it is suggested that the author double-check the small grammar errors in the full text, especially the lack and redundant use of definite articles. I will only point out some of them here, of course, not all of them. For example:
Line 16, ‘Natural resources such as coal, oil and natural gas are the natural resources that human beings have depended on for survival since ancient times.’ I suggest refining the language expression, and the bold part should not appear twice. This can directly start as, ‘Coal, oil and natural gas are the natural resources that human beings have depended on for survival since ancient times.’
Line 17, ‘By the 21st century...the total global energy consumption will increase rapidly’. Now that we have reached the 21st century, it should not be‘ will increase’, but ‘have increased rapidly’. The same applies to the latter ‘In 2021... will be around 3.5 trillion m3’. Now the data should have been fixed.
Line 71, ‘China Clean Energy’ should be ‘Clean Energy in China’, and so on.
- Line 21, ‘the coal demand will be 147.6 EJ’. The demand should be in ‘tons’ or ‘Kg’, why is the demand related to energy unit ‘J’?
- Line 73, ‘one is that clean energy is renewable energy’ and ‘the third is that clean energy refers to renewable energy or green energy’, what is the difference between them? I consider they should be merged.
- Line 146, ‘The second stage is the stage of vigorously developing clean energy, the energy structure will be transformed to green, low-carbon and intelligent, and the development of clean energy with no pollution emissions based on solar energy, wind energy, hydropower...’
- The title of the manuscript is ‘clean energy in China’. While in the whole text, only page 6 to page 8 describes solar energy, wind energy, and hydro-power. There are too many descriptions about the international background and too few details focusing on China's clean energy development. I think there can be more introductions about these three kinds of clean energy. The current manuscript is more like a popular science article than an academic paper.
- One of the biggest disadvantages of clean energy, especially the three representative energy sources mentioned in this paper, is intermittent, which is very vulnerable to natural conditions. This issue should be mentioned, and the related energy storage demand should be introduced as well. ‘Renewable energy sources such as solar energy and wind energy are unstable and intermittent during generation, and thus these valuable electric energies are difficult to apply continuously and stably. To tackle this issue, the employment of large-scale energy storage systems combined with renewable energy may greatly improve the utilization rate and stability of renewable energy [ChemSusChem 15.1 (2022): e202101798].’
- What are the limiting factors for China's clean energy development? Do national and local government care policies and subsidies contribute to the development of clean energy? The electric vehicle market, in particular, is also an important way to drive clean energy and emissions reductions, should be mentioned.
Author Response

(The authors gave the same response as above.)

Round 2
Reviewer 1 Report
Comments and suggestions for Authors:
I would like to thank authors for the response. Most of my comments were addressed which improved the paper. Please see below a few comments/suggestions to be addressed:
- I suggest changing “Gray Model” to “Grey Forecasting Model”.
- The introduction section needs to be improved with a description of the scope of work/objectives and the novelty of this paper.
- Line 13 – there is a dot before the comma. Check the entire paper for formatting issues and grammar mistakes.
- Line 35 – the sentence doesn’t make sense.
- Figure 3 and 7 quality/resolution could be improved.
- Table 2 – there is a problem with printing the table in the PDF format, see table lines.
- Equations 1-9 could be better described.
- Equation 4 – check text size.
- Table 5 - check for spelling mistakes (e.g. higer).
Author Response
Dear Reviewer:
Thank you for taking the time to review my manuscript over the holidays, and for providing us with this great opportunity to submit a revised version of our manuscript. I appreciate the detailed and constructive comments provided by the reviewers. I have carefully revised the manuscript by incorporating all the suggestions from the review panel.
I hope this revised manuscript has addressed your concerns, and look forward to hearing from you.
Yours Sincerely,
Jin Han
